# Subcutaneous Lidocaine Infusion for Chronic Widespread Pain: A Chart Review and Survey Examining the Safety and Tolerability of Treatment

**DOI:** 10.3390/jcm14072440

**Published:** 2025-04-03

**Authors:** Nina Gregoire, Kimberley Kaseweter, Ethan Klukas, Anita Sanan, W. Francois Louw

**Affiliations:** 1Department of Psychology, University of British Columbia Okanagan, Kelowna, BC V1V 1V7, Canada; kimberley.kaseweter@ubc.ca (K.K.); eklukas@student.ubc.ca (E.K.); 2Department of Anesthesiology, University of British Columbia, Vancouver, BC V6T 1Z1, Canada; anita.sanan@gmail.com; 3Department of Family Medicine, University of British Columbia, Vancouver, BC V6T 1Z1, Canada; docwyns@gmail.com

**Keywords:** lidocaine, pain management, patient survey, chronic widespread pain

## Abstract

Chronic widespread pain (CWP) is characterized by persistent pain across multiple body regions, often accompanied by fatigue, cognitive difficulties, and psychological distress. **Background/Objectives:** Affecting approximately 10% of the general population, CWP disproportionately impacts women, individuals from lower socioeconomic backgrounds, immigrants, and those with a family history of chronic pain. Standard treatments, including cognitive–behavioral therapy, exercise, and pharmacotherapy, often provide insufficient relief. This study explores a novel approach to treating treatment-resistant CWP: high-dose subcutaneous lidocaine infusions administered over extended periods. **Methods:** The research included a retrospective chart review and patient survey to evaluate safety and tolerability. The protocol started with a dose of 10–12 mg/kg of adjusted body weight, increasing by 10–15% per month, with a maximum dose of 2000 mg. **Results:** The chart review of 27 patients revealed mild to moderate adverse events (AEs) in seven patients, with no severe AEs. A survey of 15 patients indicated a higher incidence of AEs; however, all patients reported that the benefits outweighed the negatives. On average, patients experienced 61% pain relief, lasting 19 days per infusion. **Conclusions:** This study demonstrates that subcutaneous lidocaine infusions are a well-tolerated treatment for CWP, offering substantial pain relief and improving patients’ quality of life.

## 1. Introduction

Chronic widespread pain (CWP) is a debilitating condition characterized by persistent pain across multiple body regions, often accompanied by fatigue, cognitive impairments, and psychological distress [1,2]. Affecting approximately 10% of the population, CWP disproportionately impacts women and is linked to factors such as low social support, immigrant status, family history of chronic pain, manual labor, and low socioeconomic status [1,3,4,5]. Obesity, sleep disturbances, and chronic disease further increase its persistence [2]. Those meeting WP2019 criteria for CWP face an increased mortality risk, particularly among marginalized groups, highlighting the need for accessible, affordable, and effective treatments [6,7].

Determining the proportion of individuals with CWP who do not respond to treatment is challenging and inconsistently documented in large-scale epidemiological studies [8]. Clinical trials indicate that there can be high rates of treatment failure, suggesting real-world outcomes may be even more challenging [9]. Evidence of treatment failure often comes from controlled trials, such as those investigating gabapentin for neuropathic pain, where treatment may reduce pain intensity for some individuals by up to 50% [10], which suggests there is a substantial population suffering from treatment-resistant pain for whom effective pain management solutions are yet to be discovered [8].

CWP is typically best managed through a multidisciplinary stepped approach [11]. First-line interventions such as cognitive–behavioral therapy (CBT), exercise, and patient education have shown success in reducing pain sensitivity, psychological distress, and reversing deconditioning [12,13]. However, when these non-pharmacological treatments prove insufficient, pharmacotherapy is introduced. In Canada, only a limited number of medications are approved for CWP and are typically reserved for severe cases [14]. Pharmacotherapy also presents challenges, including analgesic failure, polypharmacy, adverse events (AEs), and addiction risks [9,15].

Lidocaine, first used for neuropathic pain in 1943, blocks sodium channels to reduce nociceptor sensitization and hyperexcitability [16]. Its anti-inflammatory effects reduce cytokines involved in secondary hyperalgesia and central sensitization [17,18]. Different administration methods offer advantages and limitations. Lidocaine infusions offer a minimally invasive option for chronic pain, with studies demonstrating the efficacy of intravenous (IV) lidocaine and patches for chronic and neuropathic pain [19,20,21]. Lidocaine infusions deliver a higher dose over a shorter duration, typically in line with perioperative pain management guidelines. A common protocol includes a 1.5 mg/kg bolus over 10 min, followed by continuous infusion (≤1.5 mg/kg/h) for up to 24 h [20]. This method can be effective; however, IV administration requires in-hospital monitoring due to cardiovascular risk making it resource-intensive (i.e., requires more staff and space resources) and thus potentially costly. Lidocaine patches offer a non-invasive option for pain management but may only provide modest relief. In addition, patches are primarily used for localized pain, limiting their use in treating CWP. Subcutaneous lidocaine has shown promise for cancer-related pain, though a meta-analysis found intravenous lidocaine provides only short-term relief [22,23]. While safe and potentially cost-effective, research on subcutaneous lidocaine for CWP remains limited [24,25]. Patient expectations and clinician relationships influence outcomes, emphasizing the need for patient perspectives in future research [26].

This study evaluates subcutaneous lidocaine for CWP by assessing safety and tolerability through a retrospective chart review and efficacy via a patient survey. Unlike previous studies on low-dose intravenous lidocaine (1–5 mg/kg) over short periods, this research explores higher subcutaneous doses (12–20 mg/kg) administered over longer periods (up to 24 h at 50 mg/h) [23]. The findings aim to inform clinical guidelines and optimize treatment strategies for CWP. One of the authors (WFL) has been involved in providing this form of pain management safely since 2002, out of hospital ambulatory care units in BC, Canada, with no serious (AEs) reported in this time.

## 2. Materials and Methods

Patients with CWP, including those with fibromyalgia, were referred to the lidocaine program after failing at least two mainstream interventions. These include pharmacological options such as amitriptyline, nortriptyline, duloxetine (Cymbalta), low-dose naltrexone (LDN), and targeted injections (e.g., trigger point injections, facet injections, or nerve blocks). Most patients also had received at least one manual treatment including physiotherapy, occupational therapy, hand therapy, or registered massage therapy. Many of the patients who were referred suffer from chronic neuropathic pain. Other pain conditions represented in these patients included complex regional pain syndrome (CRPS), persistent lower back pain following lumbar surgery (i.e., failed back surgery syndrome), cancer-related pain, and pelvic pain syndrome. Thus, the sample of patients consisted of a combination of refractory chronic pain patients with different etiologies. All patients paid upfront and out of pocket for their treatment; it is unknown how many had private insurance that may have covered the associated fees. All patients in the program were expected to participate in neuroeducation classes at the clinic. However, the percentage of patients who attended these classes is unknown. Needle-phobic patients were also considered candidates.

Prior to treatment, all patients required an electrocardiogram (ECG) to rule out arrhythmias or prolonged QT intervals, and they were screened for contraindications such as allergies to lidocaine, frailty, significant renal (GFR < 30 mL/min) or liver dysfunction (ALT/AST > 2.5 × normal), and cardiac pathology (e.g., heart failure, arrhythmias). Patients on medications that suppress the central nervous system (e.g., high-dose gabapentin, benzodiazepines) were given careful consideration. 

At the first visit, patients were weighed to calculate an adjusted body weight, chosen to reduce overdose risk in those with obesity. Lidocaine doses were set at 10–12 mg/kg based on this adjusted weight, calculated via an online tool [27]. This method for determining dose has been used widely within our regional healthcare system for approximately 20 years. This range has been proven to be safe and relatively well tolerated in thousands of administrations during this time. To our knowledge, no publications on this modality have emerged during this period. Patients received a detailed handout explaining the procedure, potential AEs, and post-treatment instructions. After discussing any questions, informed consent was obtained before starting treatment.

Dose increases occurred at no more than 10–15% per month if pain symptoms were not adequately treated. The ceiling dose was 20 mg per kilogram of adjusted body weight, and regardless of weight, no patient was administered more than 2000 mg lidocaine (100 mL of 2% lidocaine) per infusion. Saline (or 5% dextrose/D5W) was added in a 1:1 ratio to the 2% lidocaine. For doses less than 1500 mg (up to 75 mL 2% lidocaine), a 125 mL Avanos elastomeric pump was used with 2% lidocaine infused at a rate of 5 mL/h. For doses exceeding 1500 mg, a 270 mL Avanos elastomeric pump was used with 2% lidocaine in a 1:1 ratio with Saline or D5W to be infused at a rate of 5 mL/h (50 mg lidocaine/h). The subcutaneous line was placed in the patient’s abdominal or pectoral area and clamped off so that the patient could return home. On average, the course of infusion lasted approximately 24 h. Patients were provided with instructions to consult in the scenario that they experienced an AE. Table 1 displays the patient instructions according to the level of severity of the AE. When the infusion was complete, the patient was instructed to remove the catheter and discard the materials.

Participants had to be diagnosed with CWP and have received at least three subcutaneous lidocaine infusions. All participants were over 18 years of age. There were no exclusion criteria. Ethics approval was obtained to review 27 patient electronic medical records (EMRs). For patients who had more than three infusions, only the most recent three were analyzed. Chart reviews occurred from 26 February to 1 March 2024.

Participants were recruited using filtered EMRs. Eligible participants were then further filtered into those who consented to be emailed a survey about their treatment. Of the 27 who met the criteria, 21 participants indicated in their medical records that they would be interested in receiving emails to participate in future research hosted by the clinic. Those who agreed were emailed a letter of initial contact and a Qualtrics link to the consent form and survey. Of those who were contacted for the survey, 15 individuals completed it.

A clinic medical office assistant filtered EMRs based on inclusion criteria. The first author then collected data on the patient’s age, sex, number of lidocaine infusions, and AEs experienced during the three most recent treatments. Patients may have seen one or a combination of clinicians at the facility.

Participants were contacted via email and asked to complete a brief (i.e., approximately 10 min) online survey (Appendix A). Specifically, participants were asked to recall their experiences with their three most recent subcutaneous lidocaine infusions. The survey aimed to comprehensively understand their personal experiences with, and perspectives on, this treatment. To evaluate treatment outcomes, pain relief was evaluated using a single-item visual analogue scale (VAS), ranging from 0 to 100.

The retrospective chart review and survey data were compiled into a de-identified dataset. Survey responses were summarized using descriptive statistics (i.e., frequencies and percentages for categorical variables and means and ranges for continuous variables). All analyses were performed in SPSS v27. Responses to the final open-ended question were analyzed using conceptual content analysis to identify the impact of the treatment on participants’ lives that may not have been captured in the closed-ended questions or chart review [28,29]. Preliminary codes were derived inductively by the first author and then expanded and contracted to best fit the data.

## 3. Results

### 3.1. Chart Review

#### 3.1.1. Demographics

The chart review sample consisted of seven males and 20 females. The average age of the sample was 53, ranging from 25 to 73 years old. On average, patients had undergone 17 total infusion treatments, ranging from 3 to 29 treatments.

#### 3.1.2. Findings

Overall, 7 of the 27 patients reported AEs to their physician at their subsequent treatment session. Two patients called into the clinic to report acute nasal congestion, one of which also reported difficulty breathing. In both cases, the congestion was resolved with over-the-counter remedies (e.g., diphenhydramine). Other AEs recorded were mild in nature and are presented in Table 2. When considering the number of events per patient, of the seven who reported AEs, four reported two AEs because of treatment (e.g., headache and nausea), and three reported one AE over the course of their three most recent treatments. The remaining 20 patients did not report any AEs to their treatment provider.

### 3.2. Survey

#### 3.2.1. Demographics

The survey sample consisted of five self-reported men and ten self-reported women. The average age of the sample was 51, ranging from 25 to 73 years.

#### 3.2.2. Findings

The survey had a 100% completion rate, with all participants answering both closed- and open-ended questions. Of the 15 respondents, 14 were still undergoing treatment, while 1 had discontinued, trying a new, insurance-covered medication to reduce costs and time. In the open-ended section, two participants mentioned financial challenges, and one cited time barriers related to the treatment.

Of the 15 participants, 8 reported no AEs, and 7 reported experiencing one or more AEs (see Table 3). When asked if the benefits of the medication outweighed the AEs, all seven who experienced AEs indicated that the benefits did indeed outweigh the AEs. These findings mapped onto the open-ended survey portion, where five participants elaborated on their AEs. Specifically, one reported physical and psychological AEs (e.g., fatigue and annoyance). Likewise, four patients described having minor, manageable physical AEs (e.g., fatigue and itching, which were reduced by clamping the line) associated with treatment.

The following is a representative quote from this content: “I typically fall asleep for a few hours after the start of the infusion and am somewhat tired and groggy for six to eight hours post-infusion. This is not a negative effect and is something easily managed”. When asked about their comfort level during the treatment on a scale of 0 (not at all comfortable) to 100 (completely comfortable), respondents rated their comfort at 84, with scores ranging from 41 to 100.

#### 3.2.3. Treatment Outcomes

Participants were asked to rate how much pain relief they experienced on average from the infusion (none to maximal; 0–100%). All participants reported experiencing pain relief from the treatment. On average, they rated their pain reduction at 61%, with individual responses ranging from 13% to 93%. These results were consistent with the open-ended responses, where 10 patients elaborated on their pain reductions. For instance, one patient noted, “[Lidocaine] reduces the daily pain by about 30%, which may not sound significant, but considering my usual pain level is 8/10 all day, it’s a welcome relief”. In addition to pain reduction, five patients reported overall life improvements, three mentioned enhanced physical function and activity, two highlighted better sleep, and two described improvements in social functioning (e.g., work and family responsibilities). On average, participants reported that the relief lasted 19 days, with a range from 5 to 44 days.

## 4. Discussion

This study builds on existing evidence that lidocaine is a safe and effective treatment for various types of chronic pain [18,20,21,22]. It also extends the literature in two key ways: first, by evaluating the safety and tolerability of a novel approach involving high-dose subcutaneous lidocaine infusions over prolonged periods, and second, by capturing patient-reported experiences and perspectives. The findings indicate that the treatment was generally well tolerated, with participants reporting high levels of comfort during administration. Despite the occurrence of mild AEs in about half of the surveyed patients, all who experienced them affirmed that the benefits outweighed the negatives. Most notably, the treatment provided substantial pain relief, with participants highlighting improvements in quality of life, physical function, sleep, and social responsibilities. While a small number of participants raised challenges related to cost and time, the overall effectiveness of the treatment and the willingness of patients to continue underscores its positive impact on managing treatment-resistant CWP.

The study revealed notable discrepancies between the chart review and the follow-up survey, particularly regarding the incidence of AEs. The survey indicated a higher frequency of AEs compared to the chart documentation, which may be attributed to several factors. First, patients might have considered the AEs too mild to mention during clinical visits, deeming them insignificant. Second, healthcare providers may not have explicitly asked about certain symptoms, leading to their omission in the chart reviews. Third, we used more specific response options in the survey. This may have prompted patients to recognize and report symptoms that they may not have mentioned to a physician during or immediately after treatment.

Although the survey did not capture follow-up information on severe AEs, it is inferred that any allergic reactions were likely minor, as no patients required hospital visits, urgent care, or clinic follow-ups for serious reactions. Patients may have also failed to mention these symptoms during appointments, either because they felt the issues were trivial or simply forgot by the time of their next session.

This discrepancy underscores the value of direct patient reporting in capturing a complete picture of treatment experiences and highlights the potential underreporting of AEs in routine clinical documentation. The survey, conducted virtually at the patients’ convenience, may have given them more time to reflect and provide detailed responses, further contributing to the discrepancy.

Strengths of this study include a naturalistic examination of the population of interest (i.e., those living with CWP) via retrospective chart review followed by an in-depth examination of their perceptions of treatment. This approach provides considerable insight into this novel procedure for both clinicians treating those with CWP and persons with lived experience. Although the sample size was small, these findings, in conjunction with previous studies and common clinical practice, provide evidence that this treatment may be useful for those experiencing CWP, including those who have experienced previous treatment failure. High completion rates among participants in the survey portion enhance the reliability of the data. Further, the combination of chart reviews and follow-up surveys provides a comprehensive understanding of both clinical outcomes and patient perspectives, offering valuable insights into their opinions and experiences.

However, the study also has limitations. The small sample size, dictated by the unique nature of the population, restricts the generalizability of the results. The absence of a clinical trial design limits the ability to establish causality, and the insufficient number of patients precludes the use of advanced experimental statistics, reducing the robustness of the findings. The survey response rate may have been impacted by selection bias. Finally, our use of a retrospective patient survey may have introduced recall bias, as participants’ recollections of treatment effects could have been influenced by time and subjective interpretation.

The implications of this research are both multifaceted and significant for clinical practice and future studies. First, the findings demonstrate that subcutaneous lidocaine infusions provide substantial pain relief and improve quality of life, offering healthcare providers a viable option to recommend for chronic pain management. The high patient-reported comfort and manageable AEs further reinforce its practicality. Second, the variability in individual responses and relief duration underscores the need for personalized treatment plans and deeper exploration of factors influencing treatment efficacy. Moreover, challenges related to cost and time barriers suggest a need to enhance accessibility, possibly through better financial support from provincial funders and insurance companies. Addressing these barriers could improve patient adherence and satisfaction. Ultimately, this research highlights the importance of integrating both clinical outcomes and patient experiences in the development and implementation of effective chronic pain treatments.

For people with treatment-resistant CWP, this approach may offer much-needed relief. Unfortunately, for patients in many parts of the world, the supply fee for this treatment may not be covered by insurance. With patient relief lasting an average of 19 days per session, this potentially translates to a significant out-of-pocket expense, particularly if patients receive treatments consistently throughout the year. However, this only accounts for the direct costs of the treatment sessions themselves. Additional indirect costs, such as taking time off work to attend appointments, arranging childcare, and expenses related to travel and transportation, further increase the financial burden on patients. These factors can significantly impact access to treatment for an already vulnerable population [2,6,7].

Future research should prioritize longitudinal studies to evaluate the long-term efficacy and safety of this treatment. To improve claims around efficacy and generalizability, multiple clinics should be involved in future studies, to increase diversity in the patient sample across various settings. Comparative effectiveness research, including trials against other pain management therapies, would be valuable for determining both efficacy and cost–benefit outcomes [30]. Investigating individual variability through personalized medicine approaches, alongside comprehensive cost-benefit analyses, is essential for improving accessibility. Mechanistic studies exploring the biological underpinnings of the treatment, paired with expanded patient-reported outcomes and qualitative data, will provide deeper insights into patient experiences. Implementation research can identify optimal strategies for integrating this treatment into diverse healthcare settings, while studies on combination therapies could enhance pain relief and mitigate AEs, offering a more holistic approach to pain management.

## 5. Conclusions

This small-scale study demonstrates that high-dose subcutaneous lidocaine infusions are generally a safe and well-tolerated method to reduce pain for individuals with treatment-resistant CWP. Participants reported substantial pain relief, with improvements in quality of life, physical functioning, sleep, and social responsibilities. Some mild, manageable AEs were reported. All patients indicated that the benefits of treatment outweighed the negatives. The high levels of comfort reported during the infusions and the substantial pain reduction experienced underscore the treatment’s viability as a practical option for chronic pain management. Future research should focus on larger, longitudinal studies and explore ways to confirm treatment efficacy, ultimately ensuring that this promising intervention reaches a broader population of patients with CWP.

## Figures and Tables

**Table 1 jcm-14-02440-t001:** Standard classification of AEs and patient instructions.

Severity	Adverse Event	Instructions
Mild	Slight drowsiness or slight metallic taste	Continue infusion.
Moderate	Marked sleepiness, strong metallic taste, dizziness, ringing in the ears, numbness around mouth and tongue, vomiting	Stop the infusion for 45 min by clamping the line and reevaluate.
Severe	Moderate symptoms that persist after clamping the line *or* any of the following: change in blood pressure, irregular heart rate (heart arrhythmias), allergic reaction (hives, shortness of breath, swelling of lips or tongue), confusion, seizures	For persistent moderate symptoms, call the clinic or attend the emergency department. For all other severe symptoms, go directly to the emergency department.

**Table 2 jcm-14-02440-t002:** List of AEs endorsed from chart reviews and the number of occurrences.

Adverse Event	# of Times Endorsed
Nasal congestion	2
Initial increase in pain	2
Difficulty breathing	1
Headache	1
Nausea	1
Fatigue	1
Restlessness	1
Low mood	1
Itchy eyes	1

Note. Participants could endorse more than one AE.

**Table 3 jcm-14-02440-t003:** List of AEs endorsed from survey participants and the number of occurrences.

Adverse Event	# of Times Endorsed
Drowsy	6
Rash/Skin Irritation	4
Nausea w/o vomiting	3
Lightheaded	2
Metallic Taste	2
Swelling	2
Redness	2
Allergic Reaction	1
Confusion	1

Note. Participants could endorse more than one AE.

## Data Availability

The survey data presented in this study are available on request from the corresponding author. The data are not publicly available due to privacy restrictions.

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
