# Peer review of "Subcutaneous Lidocaine Infusion for Chronic Widespread Pain: A Chart Review and Survey Examining the Safety and Tolerability of Treatment"

_jcm, 2025, doi:10.3390/jcm14072440_

Round 1

Reviewer 1 Report

Comments and Suggestions for Authors

The research, in general, could be of some interest. The manuscript, however, should be thoroughly revised, from many points of view. Some are clearly noted in the list below. In addition to that, we believe the introduction should clarify more clearly what the authors mean by "widespread pain". This concept is not very clear, at the moment, and many readers, including me, may not understand at all what they are talking about. Many, certainly, will confuse it with what the vast majority of scientific literature today identifies as "fibromyalgia", even if the authors claim that this pathology (still poorly defined) represents only a part of their patients. Furthermore, the therapeutic modality of this "mysterious" pathological entity would make the reader wish to know more precisely what the previous treatment modalities were; and this would be great if it were reported in a specific table. As all clinicians know very well, often these poorly defined syndromes are the consequence of a fragmented and cumbersome diagnostic and therapeutic path. So, clarifying these two aspects to the readers could make the research reported in the manuscript much clearer.

In the "discussion", the manuscript also highlights aspects that are decidedly useless, such as the cost of therapy. Regardless of the fact that this aspect varies greatly, depending on the countries in which the research could be reproduced (for example, in none of the European countries with a socialized health system, this would be a problem), the topic is introduced for the first time in the "discussion", when it has not been mentioned at all either in the "methods" or in the "results". So, up to that point, that aspect is non-existent. To me, this does not seem to be a good indicator of the scientific nature of the product (manuscript) that is being presented.

It would therefore be necessary a thorough revision of the manuscript, before resubmitting it for possible acceptance.

  1. Please explain better how you decided the dosages. At the moment it is not clear to me.
  2. Please prepare table 1 in a better and more clear format. At the moment it seems confusing.
  3. Please make clear that all the patients had provided an informed consent to the treatment, and not only for the revision of their data.
  4. Please inform the readers on the scale used to evaluate the pain relief.
  5. Results 3.1 (Chart review) would be better understood in a simple table, summarizing this aspect.
  6. Conclusions should be shortened, avoiding useless repetitions

Author Response

Reviewer 1: Comments and Suggestions for Authors

Thank you very much for your insightful feedback. We have revised the manuscript to the best of our ability and considered your comments. Please find our responses below.

Comment 1: The introduction should clarify more clearly what the authors mean by "widespread pain". This concept is not very clear. Many, certainly, will confuse it with what the vast majority of scientific literature today identifies as "fibromyalgia", even if the authors claim that this pathology (still poorly defined) represents only a part of their patients. Furthermore, the therapeutic modality of this "mysterious" pathological entity would make the reader wish to know more precisely what the previous treatment modalities were; and this would be great if it were reported in a specific table. As all clinicians know very well, often these poorly defined syndromes are the consequence of a fragmented and cumbersome diagnostic and therapeutic path. So, clarifying these two aspects to the readers could make the research reported in the manuscript much clearer.

Response 1: Thank you for your comment. We have revised the introduction section and the method section to better define our sample and how it fits with the definition of CWP in the literature. 

To be clear, many of the patients that are coming in for lidocaine infusions do not have typical fibromyalgia/ CWP which is defined by “combining 4-region pain and a total pain site score ≥7” (Wolfe et al, 2019)

https://www.degruyter.com/document/doi/10.1515/sjpain-2019-0054/html

Many of our patients suffer from chronic neuropathic pain. Other pain conditions in this cohort include CRPS pain, persistent low back pain following lumbar surgery (failed back surgery syndrome (FBSS), pelvic pain syndrome and others.

Thus, we have a combination of refractory chronic pain patients in our cohort with different etiologies.

All patients receiving lidocaine infusions have failed at least two mainstream pharmacological treatments such as amitriptyline, nortriptyline, duloxetine, LDN (low-dose naltrexone), or neuropathic pain medication such as gabapentin, pregabalin (or carbamazepine for trigeminal neuralgia).

Most patients have also received at least one manual treatment including physiotherapy, occupational therapy, hand therapy, or registered massage therapy.

All patients in the program would have been expected to participate in the pain neuro-education classes at the clinic.

Comment 2: In the "discussion", the manuscript also highlights aspects that are decidedly useless, such as the cost of therapy. Regardless of the fact that this aspect varies greatly, depending on the countries in which the research could be reproduced (for example, in none of the European countries with a socialized health system, this would be a problem), the topic is introduced for the first time in the "discussion", when it has not been mentioned at all either in the "methods" or in the "results". So, up to that point, that aspect is non-existent. To me, this does not seem to be a good indicator of the scientific nature of the product (manuscript) that is being presented.

Response 2: We agree that the discussion on cost potentially has limited value for other global regions. We do acknowledge that subcutaneous lidocaine infusions have been cited by other researchers as “safe and cost effective” in the introduction section of the manuscript. This is then contrasted with the potentially costly IV lidocaine administrations that are being implemented in local hospitals (this new section was added to address other reviewer comments). Our survey “results” do indicate that one patient discontinued as a result of cost. This issue is then touched on briefly in the discussion – “a small number of participants raised challenges related to cost and time”. We have made the discussion on cost more generally and shortened it.   

Comment 3: Please explain better how you decided the dosages. It is not clear to me.

Response 3: Thank you for this suggestion. We have added a section to the manuscript describing this. The dosages have been used widely within Interior Health over approximately 20 years. This range has been proven to be safe and well tolerated in thousands of administrations during this time. Interestingly enough, nobody has published on this modality within this period.

Comment 4: Please prepare table 1 in a better and clearer format. It seems confusing.

Response 4: We added two sentences in the text to better introduce the contents of the table. We also adjusted the headings in the table to increase clarity.

Comment 5: Please make clear that all the patients had provided an informed consent to the treatment, and not only for the revision of their data.

Response 5: This has been added at bottom of page 9

Comment 6: Please inform the readers on the scale used to evaluate the pain relief.

Response 6: VAS single item for survey study was added to the method section.

Comment 7: Results 3.1 (Chart review) would be better understood in a simple table, summarizing this aspect.

Response 7: This has been added to the manuscript. We agree that including this increases the clarity of the findings. Thank you.

Comment 8: Conclusions should be shortened, avoiding useless repetitions.

Response 8: We have revised this section. Thank you.

Reviewer 2 Report

Comments and Suggestions for Authors

This manuscript presents a mixed-methods study examining the safety, tolerability, and effectiveness of high-dose subcutaneous lidocaine infusions for chronic widespread pain (CWP). The study combines a retrospective chart review of 27 patients with a follow-up survey completed by 15 patients.

Major Strengths:

  1. Novel approach: The study investigates a unique treatment protocol using higher doses of subcutaneous lidocaine over longer periods than typically studied.

  2. Mixed methodology: The combination of chart review and patient survey provides complementary perspectives on treatment outcomes and experiences.

  3. Patient-centered outcomes: The inclusion of detailed patient-reported outcomes and experiences adds valuable insight into real-world treatment effectiveness.

  4. Clear protocol description: The treatment protocol, including dosing calculations and safety procedures, is well-documented and reproducible.

Weaknesses and Suggestions for Improvement:

Methodology:

  1. The small sample size limits generalizability. The authors acknowledge this limitation but should discuss potential ways to expand the study in future research.

  2. The retrospective nature of the chart review may have introduced recall bias. This should be explicitly discussed in the limitations section.

  3. The survey response rate (15/21 contacted) should be analyzed for potential selection bias.

Results:

  1. Statistical analysis could be enhanced by including:

    • Confidence intervals for key outcomes
    • More detailed demographic analysis
    • Statistical tests for associations between patient characteristics and treatment outcomes
  2. The discrepancy between chart-documented and survey-reported side effects warrants more thorough analysis and discussion.

Discussion:

  1. The economic analysis could be expanded to include:

    • Comparison with costs of alternative treatments
    • Quality-adjusted life year (QALY) calculations
    • Discussion of potential cost-effectiveness
  2. Greater context needed regarding how these findings compare with other lidocaine administration methods (IV, patches, etc.).

Technical Points:

  1. Table 1 could be enhanced by including frequencies of observed side effects from both chart review and survey data.

  2. The methods section should clarify the time period over which charts were reviewed.

  3. Consider adding a flow diagram showing patient selection and survey response rates.

Writing and Presentation:

  1. The manuscript is generally well-written but could benefit from:
    • More consistent use of abbreviations
    • Clearer organization of the results section
    • Better integration of qualitative and quantitative findings

Author Response

Reviewer 2: Weaknesses and Suggestions for Improvement

Thank you very much for your insightful feedback. We have revised the manuscript to the best of our ability and considered your comments. Please find our responses below.

Methodology:

Comment 1: The small sample size limits generalizability. The authors acknowledge this limitation but should discuss potential ways to expand the study in future research

Response 1: Thank you for your comment. This has been addressed in the discussion, and recommendations have been added.

Comment 2: The retrospective nature of the chart review may have introduced recall bias. This should be explicitly discussed in the limitations section.

Response 2: This has been addressed in the limitations section.

Comment 3: The survey response rate (15/21 contacted) should be analyzed for potential selection bias.

Response 3: We unfortunately do not have access to sufficient data to address this. Instead, we have listed this potential bias in the limitations section. Thanks for pointing this out.  

Results:

Comment 4: Statistical analysis could be enhanced by including:

    • Confidence intervals for key outcomes
    • More detailed demographic analysis
    • Statistical tests for associations between patient characteristics and treatment outcomes

Response 4: We agree that further statistical analyses would be beneficial. Our aim is to expand on this area in future studies with larger samples. However, our ability to analyse further is limited with this data set. As such, we are not able to provide further information although we acknowledge it would be very beneficial.

Comment 5: The discrepancy between chart-documented and survey-reported side effects warrants more thorough analysis and discussion.

Response 5: We have added additional information in the discussion. The survey used more specific response options, and this may have prompted patients to recognize and report different AEs in the time after treatment compared to their in-clinic visits. This may account for the discrepancy.

Discussion:

Comment 6: The economic analysis could be expanded to include:

    • Comparison with costs of alternative treatments
    • Quality-adjusted life year (QALY) calculations
    • Discussion of potential cost-effectiveness

Response 6: We agree that these types of analyses would be helpful. Reviewer 1 felt as though we should limit our discussion on the economics of the treatment. We revised the manuscript to introduce discussion around cost of lidocaine treatments more generally. This has been included in the introduction and briefly in the results.

Comment 7: Greater context needed regarding how these findings compare with other lidocaine administration methods (IV, patches, etc.).

Response 7: This is a helpful suggestion. We have revised the introduction to include more context. IV lidocaine infusions incorporate a higher infusion rate over a shorter duration (1.5mg/kg over 10min, followed by an infusion of no more than 1.5 mg.kg/hr-for no longer than 24 hr as per peri-operative pain management guidelines (Foo, 2020).

https://associationofanaesthetists-publications.onlinelibrary.wiley.com/doi/10.1111/anae.15270

Higher infusion rates require cardiac monitoring, require more staff resources, need to be done in-hospital, and is much more costly.

Lidocaine patches are used in pain management, but these are only modestly effective and can only be used for discrete / localized pain regions.

Technical Points:

Comment 8: Table 1 could be enhanced by including frequencies of observed side effects from both chart review and survey data.

Response 8: This has been revised by adding a separate table for the chart review. Now results/frequency can be viewed separately for both the chart and survey. Thanks for this suggestion.  

Comment 9: The methods section should clarify the time over which charts were reviewed.

Response 9: This has been included. The chart reviews occurred during a 5-day period from February 26 to March 1, 2024.

Comment 10: Consider adding a flow diagram showing patient selection and survey response rates. [probably not necessary]

Writing and Presentation:

The manuscript is generally well-written but could benefit from:

Comment 11: More consistent use of abbreviations.

Response 11: This was noted for CWP and some other acronyms. Side effects and adverse events were interchanged throughout the manuscript. This has been adjusted to adverse events (AEs) throughout the article for consistency. We believe we have corrected all acronyms in the manuscript.

Comment 12: Clearer organization of the results section.

Response 12: This has been re-formatted to include the section headings and make clear that treatment outcomes section is a sub-heading under the survey results.

Comment 13: Better integration of qualitative and quantitative findings

Response 13: We have addressed the discrepancies between chart review and survey results in some more detail. A table has been added to better present the results for the chart review. We have also added some context for lidocaine administration methods that we hope improves the relevance of the current findings.

Round 2

Reviewer 1 Report

Comments and Suggestions for Authors

Thank you for thinking of this journal to submit the results of your research. The topic is not new, but the results are acceptable and worthy of being published on this high standard journal.

Reviewer 2 Report

Comments and Suggestions for Authors

No more comments to make